# GSTP1 positive prostatic adenocarcinomas are more common in Black than White men in the United States

Igor Vidal[1], Qizhi Zheng[1], Jessica L. Hicks[1], Jiayu Chen[1], Elizabeth A. Platz[2,3,4], Bruce J. Trock[2,3,5], Ibrahim Kulac[6], Javier A. Baena-Del Valle[7], Karen S. Sfanos[1,2,3,5,8], Sarah Ernst[1], Tracy Jones[1], Janielle P. Maynard[1], Stephanie A. Glavaris[5], William G. Nelson[1,2,3,5,8], Srinivasan Yegnasubramanian[1,2,3,8], Angelo M. De Marzo[1,2,3,5,8]*

1 Department of Pathology, The Johns Hopkins University School of Medicine, Baltimore, Maryland, United States of America, 2 The Sidney Kimmel Comprehensive Cancer Center at Johns Hopkins, Baltimore, Maryland, United States of America, 3 The Brady Urological Research Institute at Johns Hopkins, Baltimore, Maryland, United States of America, 4 Department of Epidemiology, The Johns Hopkins Bloomberg School of Public Health, Baltimore, Maryland, United States of America, 5 Department of Urology, The Johns Hopkins University School of Medicine, Baltimore, Maryland, United States of America, 6 Koc University, Istanbul, Turkey, 7 Fundación Santa Fe de Bogotá University Hospital, Bogotá, Colombia, 8 Department of Oncology, The Johns Hopkins University School of Medicine, Baltimore, Maryland, United States of America

* ademarz@jhmi.edu

**Data Availability Statement:** All relevant data are within the manuscript and its Supporting Information files.

## Abstract

GSTP1 is a member of the Glutathione-S-transferase (GST) family silenced by CpG island DNA hypermethylation in 90–95% of prostate cancers. However, prostate cancers expressing GSTP1 have not been well characterized. We used immunohistochemistry against GSTP1 to examine 1673 primary prostatic adenocarcinomas on tissue microarrays (TMAs) with redundant sampling from the index tumor from prostatectomies. GSTP1 protein was positive in at least one TMA core in 7.7% of cases and in all TMA cores in 4.4% of cases. The percentage of adenocarcinomas from Black patients who had any GSTP1 positive TMA cores was 14.9%, which was 2.5 times higher than the percentage from White patients (5.9%; P < 0.001). Further, the percentages of tumors from Black patients who had all TMA spots positive for GSTP1 (9.5%) was 3-fold higher than the percentage from White patients (3.2%; P<0.001). In terms of association with other molecular alterations, GSTP1 positivity was enriched in ERG positive cancers among Black men. By *in situ* hybridization, *GSTP1* mRNA expression was concordant with protein staining, supporting the lack of silencing of at least some *GSTP1* alleles in GSTP1-positive tumor cells. This is the first report revealing that GSTP1-positive prostate cancers are substantially over-represented among prostate cancers from Black compared to White men. This observation should prompt additional studies to determine whether GSTP1 positive cases represent a distinct molecular subtype of prostate cancer and whether GSTP1 expression could provide a biological underpinning for the observed disparate outcomes for Black men.

**Funding:** This work was supported by NIH/NCI SPORE in Prostate Cancer: P50CA58236, and the NIH/NCI U01 CA196390 for the Molecular and Cellular Characterization of Screen Detected Lesions (MCL), the U.S. Department of Defense Prostate Cancer Research Program (PCRP): W81XWH-18-2-0015. The Johns Hopkins Sidney Kimmel Comprehensive Cancer Center Oncology Tissue Services Laboratory supported by NIH/NCI grant P30 CA006973.

**Competing interests:** Conflicts of interest: S.Y., W. G.N., and A.M.D. are paid consultants to Cepheid LLC, with whom they are developing epigenetic tests for prostate cancer. S.Y. has received sponsored research support from Cepheid for development and testing of prostate cancer epigenetic biomarkers. This arrangement has been reviewed and approved by the Johns Hopkins University in accordance with its conflict of interest policies. This does not alter our adherence to PLOS ONE policies on sharing data and materials.

## Introduction

The pi class glutathione S transferase (encoded by the *GSTP1* gene) is a member of the cytosolic superfamily of glutathione-S-transferases. These phase II detoxification enzymes catalyze the conjugation of glutathione to diverse endogenous and exogenous electrophiles [1]. In the prostate, GSTP1 is constitutively expressed at high levels in basal epithelial cells, variably expressed in luminal epithelial cells, and epigenetically silenced in approximately 90 to 95% of adenocarcinomas by somatic hypermethylation of its regulatory CpG island [2–5]. Hypermethylation of *GSTP1* appears to occur early in prostatic carcinogenesis since it is already present in approximately 70% of high-grade PIN lesions [6,7], the presumed precursor of most invasive carcinomas, where it is silenced specifically in luminal epithelial cells [7,8]. Studies in human prostate cancer cells suggest a role for *GSTP1* as a caretaker gene whose loss increases cell survival in response to protracted oxidative injury [9]. In mouse models of carcinogen exposure induced cancers of the lung and skin, *Gstp1/2* behaves as a tumor suppressor [10,11]. In a mouse model of early prostate cancer development induced by human MYC, *Gstp1/2* also functions as a tumor suppressor [12].

GSTP1 acts as a homodimer to catalyze reactions between reactive oxygen species, anti-cancer drugs or carcinogens, and glutathione, sometimes 'detoxifying' and at other times 'toxifying' the substrates [13]. By contrast to silencing *GSTP1*, an induction of expression of GSTP1 protein is used to detect early cancer formation in response to carcinogenic exposure in the rat liver [14]. Moreover, expression and overexpression of GSTP1 are common in several human tumor types, where it has been implicated in promoting resistance to a number of chemotherapeutic agents [15]. Although GSTP1 expression has been estimated to be maintained in 5–10% of prostate cancer cases, little is known about the molecular and clinical features of these cases and only a few studies have directly asked the question of the frequency of positive GSTP1 Immunohistochemical (IHC) staining in prostate cancer [16,17].

Black men are more likely to develop prostate cancer than White men, and are more likely to have poor outcomes if diagnosed with the disease [18]. The reasons for the disparity are multifaceted, including inequities in access to high-quality prostate cancer screening, detection, diagnosis, and treatment. In addition, there appear to be at least some biological differences in disease pathogenesis and malignant progression, potentially based on associations of perceived race with genetic variation, or, as a result of different exposures. For example, when compared to prostate cancers from predominantly White men, prostate cancers from Black men contain fewer *TMPRSS2-ERG* fusions [19–22], fewer *PTEN* deletions [20–23], and more *SPOP* mutations [21,22].

In this study we comprehensively surveyed the frequency of GSTP1 protein expression in human clinical prostate cancers by IHC using tissue microarrays (TMAs). Our goal was to begin to determine whether GSTP1-positive prostate cancer represents a distinct molecular subtype, and, whether the prevalence of GSTP1-positivity differs between Black and White men.

## Materials and methods

### Study population

The study population consisted of 1673 patients who underwent radical prostatectomy (RRP) at a single center between 1993 and 2019, with an age range from 40 to 75 years old and whose prostatectomy tissue was arrayed across 45 TMA blocks. The cancers encompassed all Gleason grade groups and a spectrum of pathological stages. For IHC staining for GSTP1, we used the following TMA sets. TMA set 1 consists of prostatectomy tissue enriched for different Gleason

grade groups and pathological stages from 476 cases operated on between 1997 and 2005 (N = 11 TMA blocks). TMA set 2 consists of prostatectomy tissue from 726 patients operated on between 1993 to 2000 (N = 16 TMA blocks) [24]. TMA set 3 consists of prostatectomy tissue from 343 Black and White men, operated on between 1993 and 2019 and matched on Gleason score, pathological stage and date of surgery within 2 years (N = 9 TMA blocks) [25]. TMA set 4 is newly designed and consists of prostatectomy tissue from 353 men operated on between 2007 and 2015 from a case-cohort design (N = 9 TMA blocks; detailed design to be described in a separate publication). These TMAs were all constructed as described [24,26,27] from the index tumor (highest grade) with a 3–4 fold sampling redundancy. To compare IHC staining and the *in situ* hybridization assay for *GSTP1* mRNA, we used a novel TMA (TMA set 5), which consists of prostatectomy tissue from 31 patients operated on less than 1 year before TMA construction in which TMA slides are stored at -20˚C, which improves RNA quality for hybridizations [28]. **All clinical and pathology data used for the study were obtained from the pathology archives of the Johns Hopkins. The Johns Hopkins University School of Medicine Institutional Review Board approved this study.**

## IHC staining

We analytically validated an automated IHC assay against human GSTP1 protein using well-known cell lines and controls and human prostate cancer tissues using a mouse monoclonal antibody (Cell Signaling Technologies, #3369) at 1:400 dilution on a Ventana DISCOVERY ULTRA Autostainer using the DISCOVERY anti-HQ HRP kit as described [8]. Cell line FFPE blocks were prepared and utilized as described [29,30]. LNCaP cells were negative and DU145 and PC3 prostate cancer cell lines were positive for GSTP1 protein as expected [2]. In a subset of the above TMAs, we also performed IHC staining for ERG (rabbit recombinant monoclonal antibody; ABCAM EPR3864, Cambridge, UK) and PTEN (rabbit monoclonal; Cell Signaling Technologies, Danvers, MA), which were performed in an automated assay on the Ventana DISCOVERY ULTRA using the DISCOVERY HQ+ Amp kit. Genetically validated cell line controls were used for antibody specificity for ERG (VCAP cells are positive and PC3, LNCaP and DU145 are negative) and PTEN (PC3 and LNCaP cells are negative and DU145 cells are positive) [31,32]. Immunostaining for p63 was carried out using the mouse monoclonal antibody (clone 4A4, Biocare Medical, Concord,CA, USA) as part of a singleplex stain [33] or as part of a cocktail [34] (PIN4 staining, Biocare Medical) on the Ventana DISCOVERY ULTRA platform.

## Slide scanning, image management and scoring

Whole TMA slides were scanned on a Hamamatsu Nanozoomer, and imported into Concentriq (from Proscia). Composite images were exported from Concentriq and imported into and visualized in TMAJ [28]. For quality control of IHC staining in the TMA spots, we used the fact that a high fraction of non-neoplastic stromal cells routinely stained positively for GSTP1. We used these stromal cells as internal positive controls such that TMA cores that lacked any staining in tumor cells, and all stromal cells, were regarded as non-evaluable and were not included in the analyses. In another small subset of TMA cores, staining was very weak in intensity in both the tumor cells and stromal cells and these were also considered equivocal in terms of staining quality and were not included. Together, this poor quality staining and uninterpretable staining in weak cases comprised ~ 1% of tissue cores with carcinoma.

Scanned TMA core images were evaluated for the presence of carcinoma by two pathologists with expertise in prostate cancer (I.V. and A.M.D.). Each TMA core containing carcinoma was evaluated for GSTP1 staining positivity using a two-tiered scoring strategy in which

any tumor cell staining above background levels (nuclear, cytoplasmic, or both) were considered positive. All others were considered negative. Cases in which there was positive cancer cell staining for GSTP1 protein were categorized in two ways: any patient who had 1) any TMA spot with some positive tumor cell staining or 2) all TMA spots with some positive tumor cells staining; all other cases were considered GSTP1-negative.

In a subset of cases we also assessed ERG and PTEN to determine co-occurrence of GSTP1 positivity with ERG positivity and PTEN loss. ERG was considered positive for a given patient's tumor if any cancer cells on any TMA cores with cancer were positive and PTEN was considered lost if any TMA cores contained any tumor cells with complete PTEN absence of staining.

### *In situ* hybridization for *GSTP1* mRNA

We developed a novel *in situ* hybridization assay for GSTP1 mRNA using ACD RNAscope, which we analytically validated (Probe-Hs-GSTP1 Cat No. 453221, kit version 2.5 manual assay according to manufacturer's recommendations) using cell lines with known GSTP1 expression status for positive and negative controls as described (**S1 Fig**) [8,28]. Briefly, FFPE tissue slides were baked at 60˚C for 1 hour followed by deparaffinization in 100% xylene twice for 5 minutes each and two changes of 100% alcohol. The slides were treated with endogenous peroxidase blocking pretreatment reagent and then incubated for 15 minutes in a boiling 1× Pretreat 2 reagent (ACD) and then treated with protease digestion buffer (III, Cat. 322337) for 30 minutes at 40˚C. The slides were incubated with a RNAscope target probe as follows: for 2 hours at 40˚C, followed by signal amplification. 3,3′-Diaminobenzidine (DAB) was used for colorimetric detection for 10 minutes at room temperature. LNCaP, PC3 and DU145 prostate cancer cells were maintained as described [30]. MDA-PCA-2b cells were maintained in F-12K media containing 20% non heat-inactivated FBS, 25 ng/ml cholera toxin (Sigma cat. No. C8052), 10 ng/ml mouse Epidermal Growth Factor (Corning cat. No. 354010), 0.005 mM phosphoethanolamine (Sigma cat. No. P0503), 100 pg/ml hydrocortisone (Sigma cat. No. H0135), 45 nM sodium selenite (Sigma cat# 9133) and 0.005 mg/ml human recombinant insulin (Life Technologies cat# 12585–014) at 37˚C and 5% CO2. All cell lines were obtained from ATCC (Manassas, VA). All cell lines were authenticated using short tandem repeat profiling by the Genetic Resources Core Facility at The Johns Hopkins University School of Medicine.

### Statistical analysis

Data were tabulated and statistical tests were performed using Stata 15 for Mac OS. The primary analysis for comparisons between groups used GSTP1 protein positivity status for each patient, either any TMA core positive or all cores positive (**Tables 2–8**). Tests for differences in proportions for any TMA core or all TMA cores positive for GSTP1 protein between Black and White men, between ERG positive and negative men, and between PTEN absent and PTEN present men were performed with the Pearson Chi2 test. Tests for trends for the proportion of men who were GSTP1 protein positive across Gleason grade groups and pathological stages were performed using the Cuzick's nonparametric trend test across ordered groups.

## Results

We applied an analytically validated automated IHC assay to evaluate GSTP1 protein expression [8] in a series of TMAs constructed from patients with clinically localized primary prostatic adenocarcinomas. A total of 5,757 TMA cores containing prostatic adenocarcinoma from 1,673 patients were evaluable for GSTP1 protein scoring by IHC. These consisted of 1,197

**Table 1. Total number of patients and TMA spots with carcinoma.**

| Race | No. of patients | No. of evaluable TMA cores with carcinoma |
|---|---|---|
| Black | 336 | 1197 |
| White | 1337 | 4560 |
| **Total** | **1673** | **5757** |

cores from 336 patients that self-identified as Black (mean age = 58.4; SD 6.6) and 4,560 TMA cores from 1,337 patients that self-identified as White (mean age = 59.3; SD 6.5) (**Table 1**).

As expected, carcinoma cells in most prostatic adenocarcinomas were completely negative for GSTP1 protein (92.3%; **Table 2**). In a subset of cases, however, some or all cancer cells stained positively for GSTP1. While some cases were uniformly positive for GSTP1, in other cases containing GSTP1 positive tumor cells, we often observed heterogeneity in staining in terms of both cell number and staining intensity. This variability occurred among cancer cells at times within a given TMA core, between a man's TMA cores, and between men. **Fig 1** demonstrates representative images of these patterns of staining. Overall, 7.7% of cases contained GSTP1 positive cancer cells in a subset of TMA cores, and 4.5% had GSTP1 positive cancer cells in all TMA cores.

The percentage of adenocarcinomas from Black patients who had any GSTP1 positive cancer-containing TMA cores was 14.9%, which was 2.5 times higher than the percentage of cases with any GSTP1 positive cancer-containing cores from White patients (5.9%; P < 0.001) (**Table 2**). Further, the percentages of tumors from Black patients who had all TMA spots with cancer cells staining positive for GSTP1 (9.5%) was 3-fold higher than the percentage of tumors from White patients (3.2%; P<0.001) (**Table 2**). We confirmed that the number of evaluable TMA cores with carcinoma was similar among tumors from Black and White men, and also ran sensitivity analysis restricted to men with the same number of evaluable cores and similar results were obtained.

Cases with any or all TMA cores with cancer cells staining positive for GSTP1 occurred across all Gleason grade groups and stages, and the higher percentage of cases from Black patients positive for GSTP1 was apparent among all grade groups except in grade group 5 (**Table 3**) and except in stage group 3 (**Table 4**). In Black but not White patients, the percentage of cases with any core positive was non-significantly decreased with increasing Gleason grade group (**Table 3**) and significantly decreased with increasing pathological stage (**Table 4**).

A subset of TMAs (N = 16 TMA blocks) were stained for ERG protein by IHC, which tightly correlates with *TMPRSS2-ERG* gene fusions [35,36]. *TMPRSS2-ERG* gene fusions are the most common form of *ETS* gene family member fusions in prostate cancer and are mutually exclusive with other *ETS* family member fusions and a number of other somatic molecular alterations in prostate cancer, such as *SPOP* mutations or SPINK overexpression [37].

**Table 2. GSTP1 protein staining in cancer cells in Black and White patients.**

| | Any cores positive | | | | All cores positive | | | |
|---|---|---|---|---|---|---|---|---|
| Race | No | Yes | Total | P Value | No | Yes | Total | P Value |
| Black | 286 (85.1%) | 50 (14.9%) | 336 | | 304 (90.5%) | 32 (9.5%) | 338 | |
| White | 1,258 (94.1%) | 79 (5.9%) | 1,337 | | 1,294 (96.8%) | 43 (3.2%) | 1,339 | |
| **Total** | **1,544** | **129** | **1,673** | **< 0.001** | **1,598** | **75** | **1,673** | **< 0.001** |

N in each cell represents number of patients.

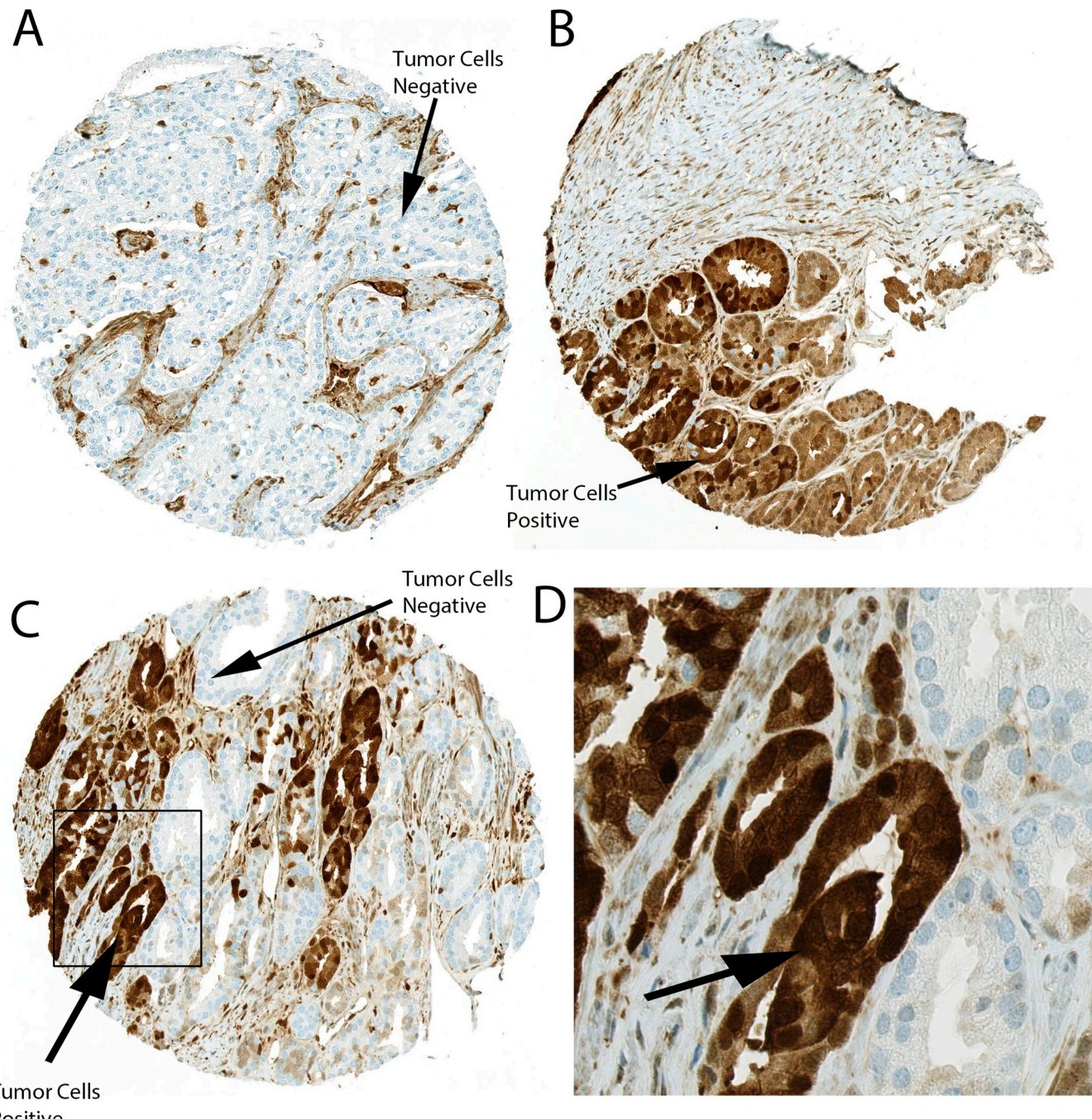

**Fig 1. Patterns of GSTP1 protein staining.** Medium power views of TMA spots containing adenocarcinoma with examples of all tumor cells being GSTP1-negative (A), all tumor cells being GSTP1-positive (B), and a spot with heterogeneous mosaic staining with some cells staining positively and some negatively (C). In all examples, there is abundant stromal cell staining that is most apparent in (A) in between tumor nests. Original magnification x 100. (D) Higher power view of boxed regions in (C).

Therefore, prostate cancers harboring *ERG* gene fusions are considered a distinct molecular subtype of prostate cancer. In this TMA subset, the percentage of tumors from Black patients with any TMA core positive for ERG was significantly lower than for tumors from White patients, consistent with prior reports [19] (**Table 6**).

**Table 3. GSTP1 protein staining by Gleason grade groups in Black and White patients.**

| | Black | | | | White | | | |
|---|---|---|---|---|---|---|---|---|
| | **Any cores positive** | | | | **Any cores positive** | | | |
| Grade group | No | Yes | Total | P Value | No | Yes | Total | P Value |
| 1 (GS 6) | 68 (77.3%) | 20 (22.7%) | 88 (100%) | | 230 (93.1%) | 17 (6.9%) | 247 (100%) | |
| 2 (GS 3+4 = 7) | 97 (86.6%) | 15 (13.4%) | 112 (100%) | | 448 (94.7) | 25 (5.3%) | 473 (100%) | |
| 3 (GS 4+3 = 7) | 43 (89.6%) | 5 (10.4%) | 48 (100%) | | 287 (95.7%) | 13 (4.3%) | 300 (100%) | |
| 4 (GS 8) | 24 (82.8%) | 5 (17.2%) | 29 (100%) | | 150 (93.2%) | 11 (6.8%) | 161 (100%) | |
| 5 (GS 9–10) | 54 (91.5%) | 5 (8.5%) | 59 (100%) | | 143 (91.7%) | 13 (8.3%) | 156 (100%) | |
| **Total** | **286 (84.9%)** | **50 (14.8%)** | **336 (100%)** | **0.039** | **1,258 (94.0%)** | **79 (5.9%)** | **1,337 (100%)** | **0.50** |
| | **All cores positive** | | | | **All cores positive** | | | |
| Grade group | No | Yes | Total | P Value | No | Yes | Total | P Value |
| 1 (GS 6) | 74 (84%) | 14 (15.9%) | 88 (100%) | | 238 (96.3%) | 9 (3.6%) | 247 (100%) | |
| 2 (GS 3+4 = 7) | 106 (94.6%) | 6 (5.4%) | 112 (100%) | | 460 (97.3%) | 13 (2.8%) | 473 (100%) | |
| 3 (GS 4+3 = 7) | 44 (91.7%) | 4 (8.3%) | 48 (100%) | | 294 (98.0%) | 6 (2%) | 300 (100%) | |
| 4 (GS 8) | 24 (82.8%) | 5 (17.2%) | 29 (100%) | | 152 (94.4%) | 9 (5.6%) | 161 (100%) | |
| 5 (GS 9–10) | 56 (94.9%) | 3 (5.1%) | 61 (100%) | | 150 (96.2%) | 6 (3.9%) | 156 (100%) | |
| **Total** | **304 (90.5%)** | **32 (9.5%)** | **338 (100%)** | **0.21** | **1294 (96.8%)** | **43 (3.2%)** | **1,337 (100%)** | **0.46** |

N in each cell represents number of patients.

When not considering race, the percentage of cases with any TMA spot or all TMA spots with cancer cells staining positive for GSTP1 was lower in those who were ERG+ than ERG- (**Table 5**). This same pattern was observed when examining White patients only (**Table 6**, any TMA spot positive). In Black patients, the opposite pattern was observed; the percentage of any TMA spot positive for GSTP1 expressing cancer was higher in men who were ERG positive than ERG negative. The percentage of patients with ERG positive cancers that were also GSTP1 positive was 23.1% in Black men, but only 5.1% in White men, a difference that was statistically significant (**Table 6**, P = 0.001). In contrast, the percentage of patients with ERG

**Table 4. GSTP1 protein staining by pathological stage in Black and White patients.**

| | Black | | | | White | | | |
|---|---|---|---|---|---|---|---|---|
| | **Any cores positive** | | | | **Any cores positive** | | | |
| Stage | No | Yes | Total | P Value | No | Yes | Total | P Value |
| 1 | 129 (78.7%) | 35 (21.3%) | 164 | | 459 (93.1%) | 34 (6.9%) | 493 | |
| 2 | 87 (89.7%) | 10 (10.3%) | 97 | | 507 (96.0%) | 21 (4.0%) | 528 | |
| 3 | 69 (93.2%) | 5 (6.8%) | 74 | | 287 (92.3%) | 24 (7.7%) | 311 | |
| **Total** | **285 (85.1%)** | **50 (14.9%)** | **335** | **<0.004** | **1253 (94%)** | **79 (5.9%)** | **1332** | **0.045** |
| | **All cores positive** | | | | **All cores positive** | | | |
| Stage | No | Yes | Total | P Value | No | Yes | Total | P Value |
| 1 | 143 (87.2%) | 21 (12.8%) | 164 | | 476 (96.6%) | 17 (3.5%) | 493 | |
| 2 | 91 (93.8%) | 6 (6.2%) | 97 | | 515 (97.5%) | 13 (2.5%) | 528 | |
| 3 | 69 (93.2%) | 5 (6.8%) | 74 | | 298 (95.8%) | 13 (4.2%) | 311 | |
| **Total** | **303 (90.5%)** | **32 (9.5%)** | **335** | **0.139** | **1289 (96.8%)** | **43 (3.2%)** | **1332** | **0.373** |

Stage groups: 1 = "Organ Confined"—T2N0; 2 –"Extraprostatic Extension"—T3aN0; 3 = Seminal Vesicle Invasion or Pelvic Lymph Node Metastasis—T3BN0 or any N1 (AJCC 2007 staging criteria). N in each cell represents number of patients.

**Table 5. GSTP1 protein staining by ERG status overall.**

| GSTP1 any TMA spot positive | ERG expression | | Total | P Value |
|---|---|---|---|---|
| | Negative | Positive | | |
| No | 238 (89.5%) | 189 (92.6%) | 427 | |
| Yes | 28 (10.5%) | 15 (7.4%) | 43 | 0.24 |
| Total | 266 | 204 | 470 | |
| GSTP1 all TMA spots positive | | | | |
| No | 247 (92.9%) | 197 (96.6%) | 444 | |
| Yes | 19 (7.1%) | 7 (3.4%) | 26 | 0.08 |
| Total | 266 | 204 | 470 | |

negative cancers that were GSTP1 positive was more similar in Black (12.9%) and White (9.4%) patients (**Table 6**, P = 0.38).

A subset of TMAs (N = 16) was also stained by IHC for PTEN, which is a known tumor suppressor in prostate cancer, whose loss is associated with disease progression [23]. Prior work has also found a lower fraction of prostatic carcinomas from Black patients with PTEN loss than carcinomas from White patients [20,21], which we observed here (22.5% PTEN loss for Black patients and 34.9% PTEN loss for White patients). When not considering race, the percentage of patients with any GSTP1-positive TMA spot appeared to be lower in men with any PTEN loss than with PTEN intact (6.6% compared with 10.3%), but this difference was not statistically significant (**Table 7**). However, stratifying by race, the percentage of any TMA spot with cancer cells staining positive for GSTP1 was comparable in those with PTEN loss and PTEN intact in Black men and in White men (**Table 8**). The percentage of patients who had both PTEN loss and were GSTP1 positive was 12.0% in Black men, but only 5.6% in White men; this difference was not statistically significant (**Table 8**, P = 0.24). These percentages were similar to those in those who had both PTEN intact and were GSTP1 positive, with 16.5% in Black men and 8.1% in White men; this difference was statistically significant (**Table 8**, P = 0.029).

Rare cases of prostate cancer have been reported that are strongly positive for nuclear p63 staining [33,38–40], and these cases tend to express GSTP1, and at times lack GSTP1 GpG island methylation [33]. We asked whether GSTP1 positive cases in the current study were ever positive for p63 nuclear staining. P63 IHC staining was available for 22 of the cases across 6 of the TMA blocks that stained positively for GSTP1 that have also been stained in our laboratory for p63; none of these cases (0/22) were positive for nuclear p63 staining. We conclude that, while many p63-positive cancers are also positive for GSTP1 [33], most GSTP1-positive cancers are not p63-positive.

To test whether samples with GSTP1 positive protein staining also contained *GSTP1* mRNA expression, we developed an ACD RNAscope *in situ* hybridization assay. We tested the

**Table 6. GSTP1 protein staining by ERG status in Black and White patients.**

| GSTP1 any TMA spot positive | ERG Positive | | | | ERG Negative | | | |
|---|---|---|---|---|---|---|---|---|
| | Race | | | | Race | | | |
| | Black | White | Total | P Value* | Black | White | Total | P Value |
| No | 20 (76.9) | 169 (94.9%) | 189 (92.6) | | 74 (87.1%) | 164 (90.6%) | 238(89.5%) | |
| Yes | 6 (23.1%) | 9 (5.1%) | 15 (7.4%) | 0.001 | 11 (12.9%) | 17 (9.4%) | 28 (10.5%) | 0.38 |
| Total | 26 | 178 | 204 | | 85 | 181 | 266 | |

**Table 7. GSTP1 protein staining by PTEN loss overall.**

| | PTEN status | | |
|---|---|---|---|
| GSTP1 any TMA spot positive | Loss (any) | Intact | Total |
| No | 141 (93.4%) | 287 (89.7%) | 428 |
| Yes | 10 (6.6%) | 33 (10.3%) | 43 |
| Total | 151 | 320 | 471 |

P = 0.19.

hybridization reaction on cell lines and found that, as expected, PC3 cells and DU145 cells were positive and LNCaP cells were negative [2] (**Fig 2**). MDA-PCa-2b cells, which were derived from an African American patient, were negative for *GSTP1* mRNA (**Fig 2**) and protein (**S1 Fig**). In normal-appearing human prostate tissue, GSTP1 protein is consistently highly expressed in basal cells, with much lower and variable expression in normal appearing luminal cells. Furthermore, the expression is highly elevated in many of the intermediate luminal cells present in the atrophic epithelium of proliferative inflammatory atrophy [7,41]. Using standard slides from RRP specimens we found a similar pattern of hybridization signals for the *GSTP1* ACD RNAscope probe set. There were also strong signals found in regions of the urethra as well as ejaculatory ducts, which is consistent with prior studies on GSTP1 protein [42] (**Fig 2**). Since RNAscope assays perform less robustly on older prostate cancer specimens [28], we performed *in situ* hybridization for *GSTP1* mRNA and IHC using a "recent case" TMA consisting of an array from 31 RRP cases (TMA set 5). We found 2 cases positive for GSTP1 protein by IHC (2/31 = 6.5%) and the same cases were also positive for *GSTP1* mRNA; all cases staining negatively for GSTP1 protein were also negative for *GSTP1* mRNA, giving a 100% concordance. We also performed IHC and *in situ* hybridization for *GSTP1* mRNA on a number of standard slides from prostatectomy specimens and found an example that was heterogeneous for GSTP1 protein in the tumor. **Fig 3** shows a region from this case that shows tight concordance between GSTP1 protein and mRNA expression.

## Discussion

In this study, we used a large number of primary untreated prostate cancer cases from prostatectomies (N = 1673) to estimate the percentage of prostatic adenocarcinomas that are positive for GSTP1 protein by IHC. We found an overall rate of any positive staining in cancer foci of 7.7%. Interestingly, we also found that GSTP1-positive carcinomas were often heterogeneous for GSTP1 protein (41%). *In situ* hybridization for *GSTP1* mRNA showed that the increased protein in the GSTP1-positive cases likely occurred through transcriptional upregulation, suggested by the tight correlation between the protein and mRNA levels. For the first time, we report that GSTP1-positive cases are significantly more common in prostatic adenocarcinomas from Black patients compared with White patients (2.5–3 fold).

**Table 8. GSTP1 protein staining by PTEN loss in Black and White patients.**

| | PTEN Loss (any) | | | | | PTEN Intact | | | |
|---|---|---|---|---|---|---|---|---|---|
| | Race | | | | | Race | | | |
| GSTP1 any positive | Black | White | Total | P Value | GSTP1 any positive | Black | White | Total | P Value |
| No | 22 (88%) | 119 (94.4%) | 141 | | No | 71 (83.5%) | 216 (91.9) | 287 | |
| Yes | 3 (12%) | 7 (5.6%) | 10 | | Yes | 14 (16.5%) | 19 (8.1%) | 33 | |
| Total | 25 | 126 | 151 | 0.24 | Total | 85 | 235 | 320 | 0.029 |

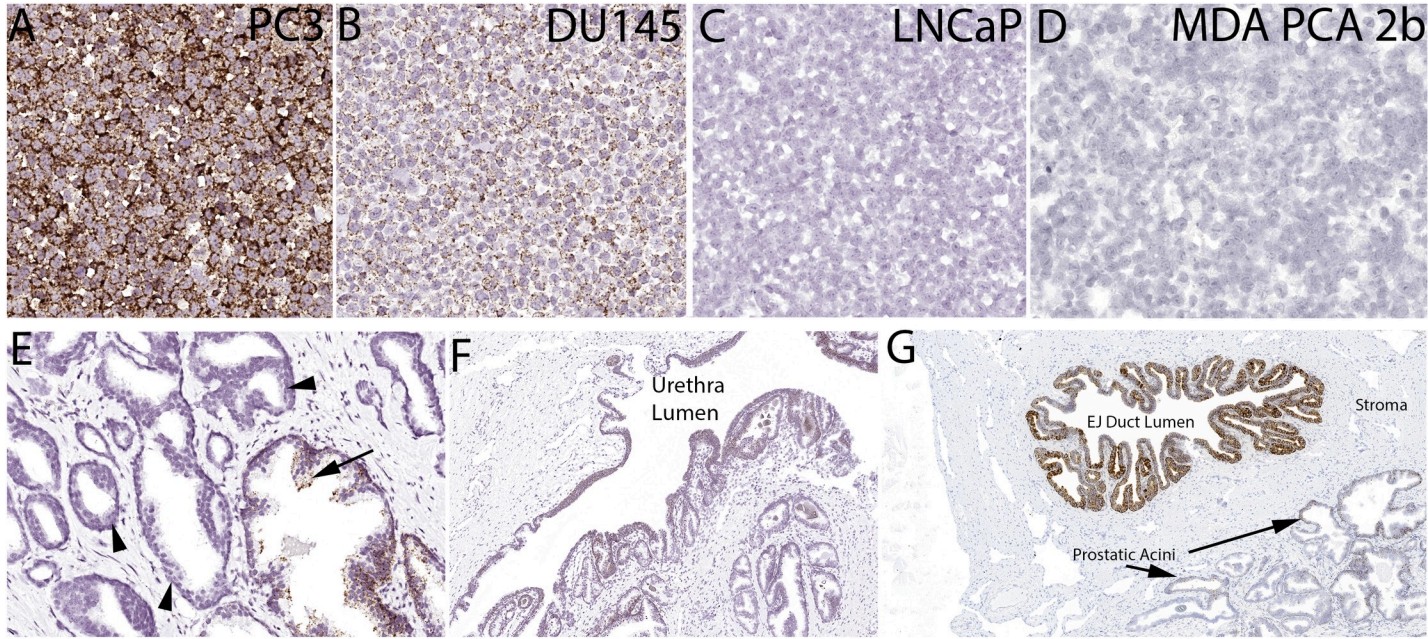

**Fig 2.** *In situ* hybridization for *GSTP1* mRNA in cell lines (A-D) and benign prostate (E-G). EJ duct lumen indicates ejaculatory duct lumen.

Given the difference in the percentage of GSTP1 positive cases by race, the current findings raise the question of whether GSTP1 positive cases represent a distinct molecular subtype of this disease. Interestingly, we found that the increase in GSTP1-positive cancers in Black men was greater in ERG-positive cases. This is of interest because prostate adenocarcinomas from Black men harbor ERG gene rearrangements less frequently than those from White men. In terms of PTEN, there was a similar increase in the percentage of GSTP1-positive cases regardless of PTEN status in Black patients.

Genomics studies of prostate cancers over the last several years have revealed the long tail of molecular alterations that can lead to cancer and influence disease progression and drug

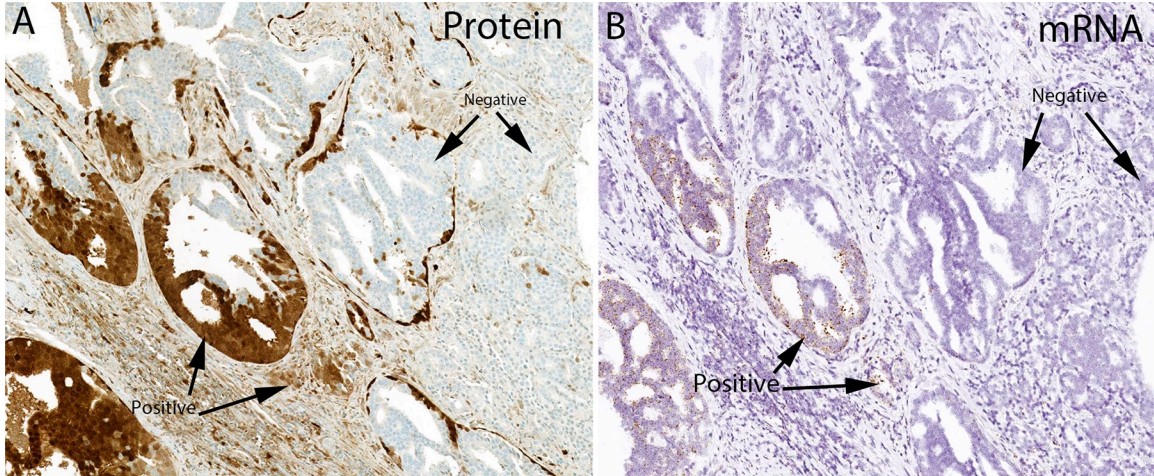

**Fig 3. Concordance between *in situ* hybridization and IHC for GSTP1 in a heterogeneous prostate cancer case.**

resistance [43–48]. These long tail events include low frequency germline and/or somatic molecular alterations involved in DNA repair (e.g., MMR defects caused by 1 of 4 different genes, *BRCA2*, *ATM* and *CDK12*). Other low frequency events considered important include *FOXA1* mutations, *IDH1/2* mutations, and mutations in a number of genes that affect epigenetic chromatin modifications. Given the large breadth of molecular alterations in prostate cancer, additional studies with larger numbers of patients are required to determine whether GSTP1 positive cases represent a distinct molecular subtype of prostate cancer.

In terms of disease aggressiveness, the percentage of men whose tumors were GSTP1- positive tended to decrease with increasing grade and stage in Black men, but not in White men (**Tables 3 and 4**). Thus, the increased frequency of GSTP1-positive cancers in Black versus White men occurred in low and intermediate grade and stage cancers only (**Tables 3 and 4**).

In terms of patient outcomes, in a series of 640 cases stained by IHC, there was no prognostic significance in univariate analysis for GSTP1 status [16]. Additional studies are underway to determine whether GSTP1 positive cases differ in terms of rates of biochemical recurrence, metastatic disease and deaths due to prostate cancer. Further, it will be of interest to determine whether GSTP1 status is related to response to therapies such as androgen deprivation, radiation, or chemotherapy and whether there are differences in the percentage of cases that are GSTP1 positive in metastatic castration sensitive and castration resistant prostate cancers. A recent study reported an association of BMI1 expression to the propensity of metastatic disease in prostate cancer in African-American men [49] and it will be interesting to correlate GSTP1 expression with that of BMI1 in future studies, including how each may contribute to the development of metastatic disease.

In terms of prostate cancer etiology, major risk factors are ancestry, family history and age. In terms of environmental exposures, at present, the fraction of prostate cancers attributable to known exposures remains quite low [50]. A limitation of our current study is that we have not determined whether men with prostate cancer that are GSTP1 positive, whether they be from Black or White men, have had different exposures than men with prostate cancers that are GSTP1 negative. Additional studies from population-based cohort studies with extensive prediagnosis exposure information are required to address this important issue and the potential racial disparity in prostate cancer burden. Also, a further limitation is that in our study we used self-identified race, which is not a biological measure; thus, future studies to include genotyping measures of percent African ancestry could perhaps help in determining whether GSTP1 expression status may be different as a result of genetic differences and/or environmental exposures or both.

To determine whether there is a different etiology for prostate cancers developing in Black versus White patients, gene expression studies can be instructive. A number of studies [21,51–53], albeit not all [54], have reported increases in inflammatory cytokine signaling in tumors from men of African descent. While inflammatory signaling and altered inflammatory cell infiltrates in the tumor microenvironment may occur during any and all temporal phases of cancer development and progression, we have previously postulated that chronic long standing inflammation may lead to prostate cancer most likely via lesions we termed proliferative inflammatory atrophy [41,55,56]. Interestingly, in regions of PIA, there appears to be an induction of GSTP1 in luminal cells, which we refer to as intermediate luminal cells [41,56,57]; with progression to PIN or adenocarcinoma accompanied by CpG island hypermethylation and loss of expression [7]. In terms of GSTP1 positive prostate cancer, it is possible that some cases arise after induction of GSTP1 in luminal cells and without silencing the gene. Martignano et al. found a strong correlation with *GSTP1* CpG island hypermethylation and GSTP1 protein status by IHC [17]. Another limitation of our study is that at present we do not know whether the expression of GSTP1 protein in the positive cases correlates with, or is

driven by, a lack of GSTP1 CpG island hypermethylation. Additional studies are underway in our laboratory to determine the CpG methylation status of *GSTP1* alleles in cases with positive staining.

Finally, the finding that GSTP1-positive prostate cancer subset is substantially over-represented among prostate cancers from Black compared to White men could be of importance from the standpoint of treatment of lethal prostate cancer, providing a potential biological underpinning, at least in part, for the observed disparate outcomes for Black men. In breast cancer, the presence or absence of GSTP1 expression predicts response to cytotoxic chemotherapy, particularly to the taxanes docetaxel and paclitaxel [58–61]. Forced expression of the enzyme in GSTP1-negative breast cancer cells confers docetaxel resistance [58]. In addition, an MCF-7 clone selected for taxane-resistance exhibited activation of GSTP1 expression [62]. Thus, the over-representation of GSTP1-positive prostate cancer in Black men may render them less responsive to taxane chemotherapy, which is the preferred chemotherapy for advanced, lethal metastatic castration resistant prostate cancer [5].

## Conclusion

In summary, we systematically examined GSTP1 protein expression in a large number of primary prostatic adenocarcinomas and found an overall rate of positivity of 7.7%, and a higher percentage of cases staining positive in Black men compared with cases from White men (2.5–3 fold). A significant difference in the percentages of GSTP1 positive cases in Black men was present only in ERG-positive cases, as compared with ERG-negative cases. These findings should stimulate additional studies regarding whether GSTP1 positive cases are a unique molecular and clinico-pathological subtype of human prostate cancer, with potential implications for disease etiology, racial disparities and therapeutic resistance mechanisms.

## Supporting information

**S1 Fig. GSTP1 IHC in prostate cancer cell lines.** IHC was performed in the cell lines indicated. PC3 and DU145 cells are positive and LNCaP and MDA-PCa-2b are negative. (TIF)

## Author Contributions

**Conceptualization:** William G. Nelson, Srinivasan Yegnasubramanian, Angelo M. De Marzo.

**Data curation:** Igor Vidal, Qizhi Zheng, Jessica L. Hicks, Jiayu Chen, Ibrahim Kulac, Javier A. Baena-Del Valle, Karen S. Sfanos, Sarah Ernst, Tracy Jones, Janielle P. Maynard, Stephanie A. Glavaris, Angelo M. De Marzo.

**Formal analysis:** Elizabeth A. Platz, Angelo M. De Marzo.

**Funding acquisition:** Angelo M. De Marzo.

**Investigation:** Igor Vidal, Qizhi Zheng, Jessica L. Hicks, Jiayu Chen, Elizabeth A. Platz, Bruce J. Trock, Ibrahim Kulac, Javier A. Baena-Del Valle, Tracy Jones, Stephanie A. Glavaris.

**Methodology:** Igor Vidal, Qizhi Zheng, Elizabeth A. Platz, Bruce J. Trock, Karen S. Sfanos, Angelo M. De Marzo.

**Project administration:** Karen S. Sfanos, Sarah Ernst, Tracy Jones, Stephanie A. Glavaris.

**Supervision:** Bruce J. Trock, Karen S. Sfanos, Angelo M. De Marzo.

**Validation:** Janielle P. Maynard.

**Writing – original draft:** Igor Vidal, William G. Nelson, Srinivasan Yegnasubramanian, Angelo M. De Marzo.

**Writing – review & editing:** Igor Vidal, Elizabeth A. Platz, Bruce J. Trock, Ibrahim Kulac, Javier A. Baena-Del Valle, Karen S. Sfanos, William G. Nelson, Srinivasan Yegnasubramanian, Angelo M. De Marzo.

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
