## [Decision Letter · Decision Letter 0]

22 Mar 2021

PONE-D-20-36882

GSTP1 positive prostatic adenocarcinomas are more common in Black than White men in the United States

PLOS ONE

Dear Dr. De Marzo,

Thank you for submitting your manuscript to PLOS ONE. After careful consideration, we feel that it has merit but does not fully meet PLOS ONE’s publication criteria as it currently stands. Therefore, we invite you to submit a revised version of the manuscript that addresses the points raised during the review process.

We look forward to receiving your revised manuscript.

Kind regards,

Craig N Robson

Academic Editor

PLOS ONE

Journal Requirements:

2. Please provide the source of the mouse tissues used in your study. Please provide details of animal ethical approval if applicable.

3. In your Methods section, please provide additional details regarding the cell lines used in your study and ensure you have described the source. In addition, please provide additional information about each of the cell lines used in this work, including any quality control testing procedures (authentication, characterisation, and mycoplasma testing).

For more information regarding PLOS' policy on materials sharing and reporting, see https://journals.plos.org/plosone/s/materials-and-software-sharing#loc-sharing-materials, and for more information on PLOS ONE's guidelines for research using cell lines, see https://journals.plos.org/plosone/s/submission-guidelines#loc-cell-lines

4. Please note that PLOS does not permit references to “[data] not shown.” Authors should provide the relevant data within the manuscript, the Supporting Information files, or in a public repository. If the data are not a core part of the research study being presented, we ask that authors remove any references to these data.

5. Please ensure your Methods and reagents are be described in sufficient detail for another researcher to reproduce the experiments described. Specifically, please provide further details on the methodology in the In Situ Hybridization for GSTP1 mRNA section.

6. In the ethics statement in the manuscript and in the online submission form, please provide additional information about the patient records/samples used in your retrospective study, including:

a) whether all data were fully anonymized before you accessed them;

b) the date range (month and year) during which patients' medical records/samples were accessed;

c) the source of the medical records/samples analyzed in this work (e.g. hospital, institution or medical center name).

7. Please ensure you have discussed any potential limitations of your study in the Discussion, including study design, sample size and/or potential confounders.

8. Thank you for stating the following in the Competing Interests section:

'Conflicts of interest: S.Y., W.G.N., and A.M.D. are paid consultants to Cepheid LLC, with whom they are developing epigenetic tests for prostate cancer. S.Y. has received sponsored research support from Cepheid for development and testing of prostate cancer epigenetic biomarkers. This arrangement has been reviewed and approved by the Johns Hopkins University in accordance with its conflict of interest policies.'

a. Please confirm that this does not alter your adherence to all PLOS ONE policies on sharing data and materials, by including the following statement: "This does not alter our adherence to  PLOS ONE policies on sharing data and materials.” (as detailed online in our guide for authors http://journals.plos.org/plosone/s/competing-interests).  If there are restrictions on sharing of data and/or materials, please state these.

Please note that we cannot proceed with consideration of your article until this information has been declared.

Additional Editor Comments:

The reviewers have requested several amendments to the manuscript and to include checking the grammar and spellings.

Any information on the methylation status of GSTP1 gene would be helpful to include and a suggestion was made to include African American cell lines for an in vitro model as important to conclude this study.

Please could you ensure that all the points raised by both reviewers have been fully addressed - this manuscript will be looked at again by the original reviewers.

Reviewers' comments:

Reviewer's Responses to Questions

**Comments to the Author**

1. Is the manuscript technically sound, and do the data support the conclusions?

Reviewer #1: Partly

Reviewer #2: Partly

2. Has the statistical analysis been performed appropriately and rigorously? 

Reviewer #1: Yes

Reviewer #2: Yes

3. Have the authors made all data underlying the findings in their manuscript fully available?

Reviewer #1: Yes

Reviewer #2: No

4. Is the manuscript presented in an intelligible fashion and written in standard English?

Reviewer #1: Yes

Reviewer #2: Yes

5. Review Comments to the Author

Reviewer #1: The paper entitled "GSTP1 positive prostatic adenocarcinomas are more common in Black than Whitemen in the United States" is interesting. However, the authors need to check and present data a little more seriously. There are many typos in the Manuscript, such as "GSPT1" as "GSP1" or White men as Whte men, etc. A little serious presentation of their findings will significantly increase the quality of the paper. I have the following comments

Comments:

1. The term "Black" and "Whitemen" are not preferable in scientific journals. I suggest the authors use the term African American and Caucasian in the "Title" and the Manuscript's main body.

2. Expand the term "TMA" in the abstract section.

3. Recently another publication observed that BMI1 expressed prominently among African Americans (PMID=30087142). How do the authors compare their findings with this publication?

4. The data on the methylation status of the GSTP1 gene will surely make the paper more valuable for the readers. Authors may include this data.

5. The number of patients between black and white patients is significantly different. Do the authors think that their difference can influence the final result of their observation?

6. Figure 2: Authors should include one or two cell lines representative of the black population. As the study emphasizes the black population, this population in vitro model is a must for this figure.

7. Figure 3: Authors must explain the background of the "representative case" in the legend. Is it from black or white patients"?

Reviewer #2: In this manuscript the authors have analyzed a large collection of prostate adenocarcinoma tissue specimen for the expression of the Glutathione-S-transferase family member GSTP1 by immunohistochemistry. Technically the analysis of the tissue micro arrays is well-performed and statistical analysis adequate. A significant higher number of GSTP1+ cases among Black men compared to White was found. However, while this finding is new and interesting, large parts of this descriptive-only study confirm already published data for GSTP1+ expression in a subset of prostate cancer patients. The authors speculate whether GSTP1 positive cancer might be a distinct molecular subtype of this disease. Molecular subtypes should be defined by genomic aberrations and/or gene expression signatures, expression of a single gene (GSTP1) is – in my opinion – not sufficient to define a molecular subtype. In addition, GSTP1+ cases were found in ERG+ and ERG- cases representing two recognized molecular subtypes of prostate cancer. To get more robust data for the definition of a molecular subtype a gene expression study could be performed on the “recent case” TMA used for RNA in situ hybridization. This would give a first indication whether GSTP1+ cases have a different gene expression signature compared to GSTP1- cases.

Minor points:

1) The GSTP1 antibody was well-controlled prior staining the TMAs. However, results of stainings in the cell lines and knock-out tissue should be added to supplementary material.

2) Which GSTP1 antibody was used? Please indicate clone, company and dilution for every antibody used.

3) Methods section: LNCaP instead of LnCAP

4) Higher magnifications of images in Figure 1 should be shown. In Fig 1C it is difficult to interprete whether the area marked with “tumor cells negative” is indeed malignant tissue?

5) Please include a table for the most important clinical parameters (age, BMI, etc) for all patients in this study (differences in mean between Black and White populations?)

6. PLOS authors have the option to publish the peer review history of their article (what does this mean?). If published, this will include your full peer review and any attached files.

Reviewer #1: **Yes: **Hifzur R Siddique

Reviewer #2: No

---

## [Author Response · Author response to Decision Letter 0]

20 May 2021

Reviewer #1: The paper entitled "GSTP1 positive prostatic adenocarcinomas are more common in Black than Whitemen in the United States" is interesting. However, the authors need to check and present data a little more seriously. There are many typos in the Manuscript, such as "GSPT1" as "GSP1" or White men as Whte men, etc. A little serious presentation of their findings will significantly increase the quality of the paper. I have the following comments

Response: We appreciate the reviewer’s positive comment that the paper is interesting. We acknowledge that there are some typos in the manuscript and we appreciate those being pointed out and have now corrected them. 

Comments:

1. The term "Black" and "Whitemen" are not preferable in scientific journals. I suggest the authors use the term African American and Caucasian in the "Title" and the Manuscript's main body.

Response: We have communicated separately with the editor about this issue, who responded to us after they communicated with the reviewer about this. Since the NCI SEER program currently prefers the terminology we used, it was agreed we would keep these designations as we have them. 

2. Expand the term "TMA" in the abstract section.

Response: We thank the reviewer for pointing that out. We have expanded the acronym TMA in the abstract. 

3. Recently another publication observed that BMI1 expressed prominently among African Americans (PMID=30087142). How do the authors compare their findings with this publication?

Response: We would like to thank the reviewer for pointing out this paper and have included a brief discussion of it in the revised submission in the discussion.

4. The data on the methylation status of the GSTP1 gene will surely make the paper more valuable for the readers. Authors may include this data.

Response: While we very much appreciate the importance of this question, we have recently begun a relatively large study of this question and plan to publish it separately as it has enough scope and complexity to present separately. The results are currently not available and are beyond the scope of the current manuscript. 

5. The number of patients between black and white patients is significantly different. Do the authors think that their difference can influence the final result of their observation?

Response: At Johns Hopkins our fraction of patients over the last several decades that are Black that undergo prostatectomy overall is 9.5% and the percentage that is White is 85.5%. In the current study we augmented these numbers by increasing the percentage of Black patients by including TMAs that were enriched for Black patients, such that our percentage of Black men in this study was 20% (see methods section for TMA dataset descriptions). We submit that our estimate found in this study of the percentage of GSTP1-positive cancers in Black men represents a highly statistically significant difference between Black and White men, and a reasonable estimate of the overall prevalence of this phenotype, since we included a relatively large number (N= 336) of Black men. To better refine our estimates, we are currently working toward increasing our overall number of both Black and White men in additional studies from patients both within our institution and beyond, although this will take more than one year at least to complete, and we feel that publishing the current findings are important to allow other researchers to also follow up on these studies

6. Figure 2: Authors should include one or two cell lines representative of the black population. As the study emphasizes the black population, this population in vitro model is a must for this figure.

Response: There are relatively few prostate cancer cell line models over all in the field (compared to that for other common cancer types), and even fewer cell lines from African American subjects. Nonetheless, we agree this is an important point and have now included the results of GSTP1 in situ hybridization from MDA-PCa-2 cell line, which was derived from an African American (PMID:9815652). We found this cell line to be negative for both GSTP1 mRNA and protein. We have revised the updated Figure 2 to include these results. 

7. Figure 3: Authors must explain the background of the "representative case" in the legend. Is it from black or white patients"?

Response: We appreciate the reviewer for pointing out that it would be helpful if provided additional information about this case. We have now added details about this case. The specimen was from a 73 year old Black man with a Gleason score of 4+5=9, and a pathological stage of T2N0Mx with margins negative. This information is now included in the manuscript.

Reviewer #2: In this manuscript the authors have analyzed a large collection of prostate adenocarcinoma tissue specimen for the expression of the Glutathione-S-transferase family member GSTP1 by immunohistochemistry. Technically the analysis of the tissue micro arrays is well-performed and statistical analysis adequate. A significant higher number of GSTP1+ cases among Black men compared to White was found. However, while this finding is new and interesting, large parts of this descriptive-only study confirm already published data for GSTP1+ expression in a subset of prostate cancer patients. The authors speculate whether GSTP1 positive cancer might be a distinct molecular subtype of this disease. Molecular subtypes should be defined by genomic aberrations and/or gene expression signatures, expression of a single gene (GSTP1) is – in my opinion – not sufficient to define a molecular subtype. In addition, GSTP1+ cases were found in ERG+ and ERG- cases representing two recognized molecular subtypes of prostate cancer. To get more robust data for the definition of a molecular subtype a gene expression study could be performed on the “recent case” TMA used for RNA in situ hybridization. This would give a first indication whether GSTP1+ cases have a different gene expression signature compared to GSTP1- cases.

Response: We appreciate the positive feedback regarding the performance of GSTP1 IHC and analysis on a large collection of adenocarcinoma specimens. 

In terms of defining a new molecular subtype, we agree with the Reviewer that the expression of a single gene, such as GSTP1, is not sufficient to define a molecular subtype. In a number of locations in the manuscript we indicate that GSTP1 status may help define a novel molecular subtype, not that it indeed does define such a subtype. For example, in the abstract we stated: “This observation should prompt additional studies to determine whether GSTP1 positive cases represent a distinct molecular subtype of prostate cancer.” Also, in the discussion we stated: “Given the large breadth of molecular alterations in prostate cancer, additional studies with larger numbers of patients are required to determine whether GSTP1 positive cases represent a distinct molecular subtype of prostate cancer.”

We further agree that it is important to obtain more robust data for the identification of a molecular subtype by performing a gene expression study. In fact, we are currently performing such a study on a large number of fresh frozen tissues in which we have GSTP1 expression and RNAseq. The data from that study are still being analyzed and will take some time to complete given the substantial scope of that effort. Thus, that analysis is beyond the scope of the current study and we submit that the current study will also prompt others to perform similar investigations. 

Minor points:

1) The GSTP1 antibody was well-controlled prior staining the TMAs. However, results of stainings in the cell lines and knock-out tissue should be added to supplementary material.

Response: We thank the reviewer for this suggestion. We have added a supplemental Figure (supplemental Fig. 1) to show the cell lines for both IHC and in situ hybridization. We have removed the reference to the knockout tissue as we realized those were stained with a different antibody. 

2) Which GSTP1 antibody was used? Please indicate clone, company and dilution for every antibody used.

Response: We appreciate pointing out this omission and have added the details for this.

3) Methods section: LNCaP instead of LnCAP

Response: We have fixed this typo.

4) Higher magnifications of images in Figure 1 should be shown. In Fig 1C it is difficult to interprete whether the area marked with “tumor cells negative” is indeed malignant tissue?

Response: We have added a higher power image as a separate panel to the revised Figure 1.

5) Please include a table for the most important clinical parameters (age, BMI, etc) for all patients in this study (differences in mean between Black and White populations?) 

Response: At present, although we don't have BMI on all of these patients, we will annotate that information for a followup study when we look at patient outcomes with respect to GSTP1 expression status. We did add the mean and SD of the patient ages by race in the results section, first paragraph in the revised manuscript.

---

## [Editor Report · Decision Letter 1]

2 Jun 2021

GSTP1 positive prostatic adenocarcinomas are more common in Black than White men in the United States

PONE-D-20-36882R1

Dear Dr. De Marzo,

We’re pleased to inform you that your manuscript has been judged scientifically suitable for publication and will be formally accepted for publication once it meets all outstanding technical requirements.

Kind regards,

Craig N Robson

Academic Editor

PLOS ONE

Additional Editor Comments (optional):

The authors have fully addressed all the comments raised by both reviewers. The manuscript has been improved by inclusion of additional data and through individual amendments to the main text as suggested by the reviewers.
---

## [Editor Report · Acceptance letter]

17 Jun 2021

PONE-D-20-36882R1 

GSTP1 positive prostatic adenocarcinomas are more common in Black than White men in the United States 

Dear Dr. De Marzo:

I'm pleased to inform you that your manuscript has been deemed suitable for publication in PLOS ONE. Congratulations! Your manuscript is now with our production department. 

Kind regards, 

on behalf of

Prof Craig N Robson 

Academic Editor

PLOS ONE